# Factors Influencing Polish Women’s Preference for the Mode of Delivery and Shared-Decision Making: Has Anything Changed over the Last Decade?

**DOI:** 10.3390/medicina58121782

**Published:** 2022-12-03

**Authors:** Maciej Walędziak, Agnieszka Jodzis, Anna Różańska-Walędziak

**Affiliations:** 1Department of General, Oncological, Metabolic and Thoracic Surgery, Military Institute of Medicine—National Research Institute, Szaserów 128 St., 04-141 Warsaw, Poland; 22nd Department of Obstetrics and Gynecology, Medical University of Warsaw, Karowa 2 St., 00-312 Warsaw, Poland; 3Department of Human Physiology and Pathophysiology, Faculty of Medicine, Collegium Medicum, Cardinal Stefan Wyszynski University in Warsaw, 01-938 Warsaw, Poland

**Keywords:** cesarean delivery, vaginal delivery, shared decision-making, cesarean delivery on maternal request, maternal preference

## Abstract

*Background and Objectives:* Shared-decision making has become an important trend in the problem of women’s preference for the way of delivery. There are different factors influencing women, including obstetric history, culture, religion, family and social influences. *Materials and Methods:* The study was designed as an online survey with the aim of acquiring information about women’s knowledge, opinions and preferences about the mode of delivery and the decision-making process. Data were collected from 1175 women in 2010 and 1033 in 2020. *Results:* A significant increase in the proportion of women who prefer vaginal delivery (VD) was found to be present with an increasing level of education, with the lowest rate in the group with primary education (66.0% in 2010 and 33.3% in 2020) and highest with medical education—86.3% in 2010 and 69.3% in 2020 (*p* < 0.05). This trend existed both in 2010 and 2020; however, the proportion of women who preferred VD has decreased over the last decade in all groups, and even two-fold in the primary education group. No significant correlation was found between a history of previous delivery and the preference of the way of delivery, decision-making or paid cesarean delivery on maternal request (CDMR). A history of VD significantly reduced the preference for having a cesarean delivery, with only 6.9% of women in 2010 with a history of VD, and 8.9% in 2020 having preferred a cesarean delivery. In 2010, 34.9% of women with a history of cesarean section (CS) only, compared to 6.9% of women with a history VD only, had preference for CS with, respectively, 36.4% vs. 5.8% in 2020. *Conclusions:* As the proportion of women who prefer cesarean delivery has significantly increased over the last decade, we should emphasize the importance of educating women about the advantages and disadvantages of vaginal and cesarean delivery. The patient’s preference should always be discussed with the obstetrician and the medical indications explained.

## 1. Introduction

The decision about the optimum mode of delivery is always very important and sometimes may be difficult. There is a strong trend towards increasing the patient’s role in the decision-making process, which allows patients take an active part in their treatment [1]. Even though maternal preference may not always be the optimum choice for the infant’s health and well-being, a mismatch between maternal preference for cesarean section and having vaginal childbirth may lead to postpartum depression and increased post-traumatic stress symptoms [2,3]. Therefore, the mother’s preference has to be taken into consideration in the decision-making process, although it is the obstetrician who should have the final word, as cesarean delivery is an operative procedure with possible negative consequences for the mother and the baby [4]. There are different factors influencing maternal preference for the mode of delivery, including age, education, relatives and friends, socioeconomic status, obstetric history, assisted reproduction techniques, nationality, culture and religion [5,6,7]. However, the most common reasons for women’s choice of cesarean delivery (CS) are fear of childbirth and lack of control, followed by concern for fetal health and fear of pelvic floor damage [5,8,9,10]. The importance of reliable information about the advantages and disadvantages of different ways of delivery and possibilities of pharmacological and non-pharmacological methods of pain relief in labor, given by healthcare professionals during pregnancy, are of utmost importance [11,12,13]. There are differences in the attitudes of national obstetrical societies about cesarean delivery on maternal request (CDMR); even though some accept CDMR, most unanimously emphasize the final and decisive role of the obstetrician [14]. The level of acceptance of CDMR differs between obstetricians in different countries, from 84.5% of Maine members of American College of Obstetricians and Gynecologists, 77.3% of obstetricians in Australia to 10% in Canada, 15% in Spain and 14.3% in China [4,15]. The preference of the mode of delivery also differs between countries, from 3.1% of women declaring preference for CS in United States to 34.1% in Iran [16,17]. Therefore, the primary objective of this study was to find factors influencing women’s preference of the mode of delivery. The secondary purpose of the study was to analyze and compare women’s present opinions and those from the previous decade.

## 2. Materials and Methods

The study was designed as an anonymous online and paper survey with the aim of collecting data about Polish women’s knowledge, opinion and preference of different ways of delivery. This study was a part of a bigger project that also included data about women’s knowledge about cesarean deliveries and pharmacological and non-pharmacological methods of pain relief in labor that have been presented in our other manuscripts. The survey was conducted in 2010 and 2020, with the same questions at both time-points, and the online version distributed via social media. Data were collected from 1175 women in 2010 and 1033 in 2020. The questionnaire included questions about the basic characteristics of the respondents (age, level of education, socioeconomic status, place of residence, presence of comorbidities), obstetric history (abortions, vaginal deliveries (VD), difficult deliveries, CS) and their knowledge, preference and opinion about different ways of delivery, CDMR and the decision-making about mode of delivery. The exclusion criteria were non-female gender, minority (less than 18 years old) and missing or conflicting data.

### 2.1. Statistical Analysis

Statistical analysis was performed using Statistica 13 (StatSoft. Inc., Tulsa, OK, USA) The U-Mann Whitney test and Student’s t tests were used for quantitative data comparison as required. The two-sided Fisher’s exact test and chi-square test were used for categorical and binary data comparison as required. A *p* value < 0.05 was considered significant.

### 2.2. Ethical Considerations

The study was anonymous, and performed in accordance with the ethical standards described in the 1964 Declaration of Helsinki and its later amendments (Fortaleza). Participants were informed about the aim of the study, and informed consent was obtained electronically prior to the beginning of the survey. Approval from Warsaw Medical University Ethics Committee was obtained was obtained on 19 March 2013 with code AKBE/21/13.

## 3. Results

The vast majority of both groups of respondents was of reproductive age—95% in 2010 and 98% in 2020. The medium age of the 2010 group was 28.0 (SD 8.8) compared to 32.0 in 2020 (SD 6.7). Most women who filled in the survey lived in cities of more than 50,000 inhabitants (68% at both time-points): 10.0% participants in 2010 and 8.2% in 2020 declared medical education, with 49.5% and 63.7% having higher education, respectively. The great majority of both groups had socioeconomic status of medium or higher level (93.7% vs. 98.6%). The baseline characteristics of the groups are presented in Table 1.

### 3.1. Educational Level

In 2010 4.3% of respondents declared primary education, 34.9% secondary, 48.8% higher and 12.1% medical education, with 1.0%, 26.7%, 62.3% and 10.1% in 2020, respectively.

In 2010, VD was preferred by 66.0% of women with primary education, 69.0% with secondary education, 70.4% with higher education and 86.3% of those with medical education (*p* < 0.05). In 2020, VD was preferred by 33.3% of women with primary education, 57.1% secondary, 55.3% higher, and 69.3% medical education (*p* < 0.05).

Women were also asked about the decision-making process concerning the mode of delivery. Among the respondents with primary education, 18.9% in 2010 and 52.6% in 2020 (*p* < 0.05) thought women should have the independent right to decide about the mode of delivery, 58.5% vs. 26.3% (*p* < 0.05) preferred shared decision-making with their obstetrician, and 22.6% vs. 21.1% accepted CS only for medical indications (*p* < 0.05). The independent right to decide about the mode of delivery was recognized by 32.1% in 2010 and 33.0% in 2020 (*p* < 0.05) of women with secondary education, 26.3% vs. 36.4% (*p* < 0.05) with higher education and 10.8% vs. 26.4% with medical education (*p* < 0.05). Shared decision-making was the preference of 46.0% in 2010 and 44.0% of women in 2020 with secondary education, 46.8% vs. 46.2% with higher education and 38.7% with medical education (*p* < 0.05). CS for medical indications only was chosen by 21.0% of women with secondary education in 2010 vs. 20.6% (*p* < 0.05) in 2020, 24.6% vs. 16.4% (*p* < 0.05) with higher education and 50.5% vs. 36.8% with medical education (*p* < 0.05).

In 2010, 18.9% of women with primary education were willing to have CDMR, with respectively 27.4% with secondary, 23.1% with higher and 13.8% with medical education (*p* < 0.05). Results from the 2020 group did not have statistical significance. In 2010, 37.8% of women with primary education were willing to pay for CDMR, compared to 15.8% in 2020, respectively; 40.1% vs. 36.7% with secondary education, 45.1% vs. 51.2% with higher education and 27.2% vs. 39.1% with medical education (*p* < 0.05 for all subgroups in 2010).

### 3.2. Medical Education

When respondents were divided into medical education and other education groups, there were significant differences found in their opinions about almost all factors concerning the mode of delivery. Both in 2010 and 2020, less women with medical education considered CS a better way of delivery, respectively, 10.3% vs. 18.2% in 2010 (*p* < 0.05) and 18.2% vs. 24.9% in 2020 (*p* < 0.05). In 2010, 50.5% of women with medical education thought that CS should be performed only in case of medical indications, compared to 23.0% of the other educational group (*p* < 0.05); respectively, 36.8% vs. 17.7% in 2020 (*p* < 0.05). Only 13.8% of women with medical education decided to have CDMR in 2010, i.e.,24.6% with other education (*p* < 0.05), whereas in 2020 the proportion was similar in both educational groups, i.e., 33.3% vs. 34.5%. In 2010, 27.2% of women with medical education accepted the possibility of having paid CDMR, compared to 42.7% of women with other education; respectively, 39.1% vs. 46.3% in 2020 (*p* < 0.05).

### 3.3. Socioeconomic Status

Socioeconomic status declared by the respondents was found to have no correlation with the preference of the mode of delivery, opinion about the level of women’s independence in the decision-making process, or CDMR. In 2010, the proportion of women who accepted paid CDMR increased with socioeconomic status, with 31.5% in the low-income group, 40.7% in medium income and 49.7% in the high-income group (*p* < 0.05); such a correlation did not exist in 2020.

### 3.4. Comorbidities, Present Pregnancy, Miscarriage

A total of 75.0% of respondents in 2010 and 71.4% (*p* = 0.34) in 2020 declared no comorbidities. At the time of filling in the questionnaire, 18.25% of women in 2010 and 9.03% (*p* = 0.13) in 2020 were pregnant, while 15.8% of women in 2010 had a history of miscarriage vs. 20.0% (*p* = 0.60) of women in 2020. Suffering from comorbidities, being pregnant at the moment of filling in the survey, or a history of miscarriage, did not influence the preference of the way of delivery, the opinion about the decision-making process, CDMR or paid CDMR. The results are presented in Table 2.

### 3.5. History of Delivery

A total of 51.4% of respondents in 2010 had a history of previous pregnancy vs. 77.1% in 2020 (*p* < 0.05). No significant correlation was found between a history of delivery (with no differentiation for vaginal and cesarean delivery) and the preference of mode of delivery, decision-making or paid CDMR. In 2010, patients with a history of delivery were more willing to have CDMR than those who had no deliveries, 20.9% vs. 26.9% (*p* < 0.05), compared to 2020 when the proportion was the same in both groups, at 34.3%.

When we compared women who had had only a VD with those who had never had deliveries, we found that in 2010 only 6.9% of those who had had VD preferred CS as a better way of delivery compared to 18.7% of nulliparas (*p* < 0.05); respectively, in 2020 8.9% vs. 25.3% (*p* < 0.05). In 2010, 18.6% of women who had a VD thought that it should be a woman’s autonomic right to decide about the mode of delivery in comparison with 29.9% (*p* < 0.05) of nulliparas; 43.8% vs. 49.7% (*p* < 0.05) opted for shared decision-making with the obstetrician and 34.5% vs. 19.8% (*p* < 0.05) stated that CS should be performed only for medical indications; respectively, 26.8% vs. 35.0%, 44.8% vs. 47.5% and 27.4% vs.16.0% in 2020 (*p* < 0.05). In 2010, 26.9% of nulliparas wanted to have CDMR, compared to 12.4% (*p* < 0.05) of women with a history of VD, and 34.4% vs. 19.1% (*p* < 0.05) in 2020. A history of VD was not found to influence the acceptance of paid CDMR.

In 2010, 18.6% of nulliparas compared to 35.2% of women with a history of CS indicated CS as their preferred way of delivery (*p* < 0.05), whereas in 2020 the proportion of nulliparas who chose CS increased to 25.3%, while the proportion of women with a history CS remained at a similar level of 36.7% (*p* < 0.05). A history of CS did not influence the opinion about decision-making of paid CDMR. As to CDMR in general, women with a history of CS were more probable to prefer CDMR, with 38.2% in 2010 and 47.3% in 2020, compared to 26.9% of nulliparas in 2010 and 34.4% in 2020 (*p* < 0.05).

In 2010, 34.9% of women with a history of CS only, compared to 6.9% of women with a history VD only, had preference for CS with, respectively, 36.4% vs. 5.8% in 2020 (*p* < 0.05). In the CS group, 46.1% in 2010 and 37.8% in 2020 preferred VD, compared to 87.5% in 2010 and 85.0% in 2020 in the VD group (*p* < 0.05). Opinions about shared decision-making did not significantly differ between CS and VD groups, with a higher preference for independent women’s decision in the CS group both in 2010 and 2020; respectively, 34.0% vs.18.6% in 2010 and 41.5% vs. 24.3% in 2020 (*p* < 0.05). Out of CS group, 38.2% in 2010 and 47.1% in 2020 would prefer CDMR, compared to 12.2% in 2010 and 15.2% in 2020 of VD group (*p* < 0.05).

Among women who had a VD, 42.4% in 2010 and 57.7% in 2020 declared it was a difficult delivery (*p* < 0.05). A history of a difficult delivery was found to have no statistically significant influence on the women’s opinion whether VD or CS was a better way of delivery. However, 22.5% of women in 2010 with a history of difficult delivery would have decided to have a CMDR compared to 9.3% of women without such a history and, respectively, 32.4% vs. 19.2% in 2020 (*p* < 0.05). 

## 4. Discussion

We found a correlation between the level of education and the preferred mode of delivery, with a significant increase in the proportion of women who prefer VD with the increasing level of education, with the lowest rate in the group with primary education (66.0% in 2010 and 33.3% in 2020) and highest with medical education, i.e., 86.3% in 2010 and 69.3% in 2020 (*p* < 0.05). This trend existed both in 2010 and 2020; however, the proportion of women who preferred VD has decreased over the last decade in all groups, and up to two-fold in the primary education group.

A similar correlation was present between the level of education and the decision-making process about the mode of delivery, with the higher proportion of women who accepted the role of obstetrician and preferred shared decision-making and understood the importance of medical indication in the higher and medical education groups. Between 2010 and 2020, there was an increase in all educational level groups, with almost a three-fold increase in the rate of women with lower education who wanted to have an independent right to choose the mode of delivery. A decrease in all educational level groups was observed in terms of preference for CS only for medical indications.

A history of VD significantly reduced the preference for having a cesarean delivery, with only 6.9% of women in 2010 with a history of VD and 8.9% in 2020 having preferred a cesarean delivery. A history of VD also influenced the opinion about CDMR; in both 2010 and 2020 groups the proportion of women who were interested in CDMR was almost two-fold lower in the group with a history of VD. A history of CS increased the preference for operational delivery, as the proportion was almost two-fold higher in the group with a history of CS (compared to nulliparas) in 2010 and a half higher in 2020, due to the trend of increasing preference for CS in the whole group. The influence of the history of VD or CS on the women’s preference was the most visible when we compared the group with a history of only VD vs. those with a history of only CS, with a higher preference for the known way of delivery. The proportion of women who had had only a CS and preferred CS was almost six times higher than of those with a history of VD only both in the 2010 and 2020 groups. The preference for VD was two-fold higher in the group with a history of VD compared to CS. A history of CS also resulted in almost two-fold higher proportion of women who thought it should be their autonomic right to decide about the mode of delivery, and indicated they would like to have a CDMR.

In a Swedish study by Kalström et al. with a group of 693 women, women who preferred and actually delivered by CS experienced a higher level of fear of childbirth than those who preferred and had VD. Women who had a CDMR were less satisfied with antenatal care and had a more negative birth experience. Swedish findings, even though focused on a slightly different aim, were only partly similar to ours. They also found that lower education was correlated with a higher rate of preference for CS. However, they found that CS was often perceived as a negative event and, in some women, resulted in a decision of not having more children, whereas in our survey a history of CS increased the preference for operational delivery in future pregnancies [8].

In a study by Fobelets et al, 862 women after one cesarean section from Germany, Ireland and Italy filled in surveys about their preference for the mode of delivery. The researchers analyzed the influence of the relation between the preference and the actual way of delivery on the postnatal Health Related Quality of Life (HRQoL). The highest HRQoL was found in the group of women who preferred and had vaginal birth after cesarean delivery (VBAC). The lowest HRQoL was in case of preference for VBAC and actual birth by an elective CS. In the presented study, 23.2% of participants preferred CS, whereas in our study 35.2% of women with a history of CS in 2010 and 36.7% in 2020 would have liked to have a CS next [18].

In a study by Kjerulff et al., 3006 nulliparous women were asked during their first pregnancy about their preferred mode of delivery. The period of gathering data was between 2009 and 2011, corresponding with the first stage of our study. Kjerulff found that only 3.1% of women preferred CS as the mode of delivery. The level of preference for CS among Polish nulliparous women was higher than in the American study at the first stage of our study, with a further risen over the last decade. We found that both in 2010 and 2020, women who had never had a delivery, via VD or CS, had a higher rate of preference for CS; respectively, 18.7% in 2010 and 25.3% in 2020. However, we have to take into consideration that women in the Kjerulff’s study were pregnant at the time of filling in the survey, whereas in our study the majority of the participants were not pregnant at the time [16].

There are numerous factors influencing women’s preference for the mode of delivery. In our study, the significant factors were the level of education, including medical education, and the obstetric history. Suffering from comorbidities, being pregnant at the time of filling in the survey, or a history of miscarriage, did not influence the preference of the mode of delivery, the opinion about the decision-making process, CDMR or paid CDMR.

The factors influencing women’s preference for the mode of delivery differs between the countries. In a study by Rajabi et al. in a group of 2191 women in Iran, 748 had a preference for CS. The factors found to have had influence on their decision were age, educational level of their spouses, number of live births and preconceived maternal attitudes about delivery [17]. In an Ethiopian study by Tenaw et al., factors that influenced women’s preference for CS were previous pregnancy complications and no cardiotocography available during the delivery. The Ethiopian study group included 300 women who had delivery in city hospitals, which may have influenced the structure of the group, as the hospitals were places of higher reference for women with obstetric complications [19]. In a Chinese study by Liang et al., the strongest factors for preferring CS were choosing a lucky day for baby birth, age of 40 years old and above, ethnic minority, difficulties in getting pregnant, and the husband’s preference for CS. The main reasons for women’s preference for CS were belief in higher safety of CS than VD for both mother and baby and belief that CS was associated a lower level of pain [20]. A study by Preis et al. in a cohort of 832 primiparous women in Israel indicated that being religious, and therefore having a preference for a higher number of children, was the main factor including the preference for VD in the study group. Women who had a history of VD, and those who believed that birth is a medical process, were more likely to have chosen CD (cesarean delivery). Factors influencing preference for VD were being more religious, higher education, spontaneous conception, a history of CD, and perception of better treatment from the medical staff [21]. Løvåsmoen et al. analyzed a cohort of 2177 pregnant women, 3.5% out of primiparous and 9.6% of multiparous having preferred CS to VD. The main reasons for CS preference were fear of childbirth, lower educational level, symptoms of depression, and age over 35 years in the primiparous group, and/or negative birth experience among the multiparous [22].

In our study, a history of difficult delivery was not a statistically significant factor for choosing either CS or VD, but increased the preference for CDMR. In a Norwegian study by Gaudernack et al., a history of prolonged labor more than doubled the risk (OR 2,66, 95%CI 1.42–4.99) of a subsequent wish for CS [23].

### Limitations of the Study

A possible limitation of our study was recall bias and the subjectivity of patients’ opinion. Another limitation was that the survey was conducted mostly among women who were able to complete it by means of the internet, and, therefore, distribution via social media excluded the possibility of direct control of the respondents or calculation of the response rate. However, there was no incentive to introduce dishonesty into responses.

## 5. Conclusions

The proportion of women who prefer cesarean delivery has significantly increased over the last decade. This alarming trend indicates we should emphasize the importance of educating women about the advantages and disadvantages of vaginal and cesarean delivery. Medical professionals, such as obstetricians and midwives involved in the pregnancy care should provide women adequate information about the ways of delivery in the course of pregnancy so that they are aware of the indications for cesarean delivery and its possible consequences for the mother and baby. Some of the main factors influencing the preference for cesarean delivery, such as fear of childbirth or concern for fetal health, are modifiable, and thorough educational and psychological preparation would decrease the increasing predominance of CS preference. Over the last decade, we have observed an increasing rate of women who prefer a shared decision-making model; therefore, the patient’s preference should always be discussed with the obstetrician, and the medical indications explained.

## Figures and Tables

**Table 1 medicina-58-01782-t001:** Basic characteristics of subjects.

Characteristics			Year 2010 *n* = 1175	Year 2020 *n* = 1033	*p*-Value
Age			28.0 (±8.8)	32.0 (±6.7)	<0.05
Socioeconomic status-income	Low		73 (6.3%)	15 (1.4%)	<0.05
	Medium		944 (81.0%)	792 (74.0%)	
	High		148 (12.7%)	264 (24.7%)	
Education	Primary		53 (4.5%)	19 (1.8%)	<0.05
	Secondary		425 (36.1%)	283 (26.4%)	
	Higher		583 (49.5%)	683 (63.7%)	
Medical education			118 (10.0%)	88 (8.2%)	
Place of habitation:		Cities > 100,000	663 (56.2%)	613 (57.1%)	0.945
		Cities 50,000–100,000	139 (11.8%)	120 (11.2%)	
		Cities < 50,000	166 (14.1%)	146 (13.6%)	
		Village	211 (17.9%)	195 (18.2%)	
No comorbidities			828 (75.3%)	725 (71.2%)	<0.05
Nullipara			573 (48.6%)	246 (22.9%)	<0.05
Ongoing pregnancy			215 (18.25%)	97 (9.03%)	<0.05
History of miscarriage			178 (15.2%)	215 (20.8%)	<0.05
History of vaginal delivery			351 (31.2%)	374 (34.8%)	<0.05
History of cesarean section			179 (15.9%)	436 (40.5%)	<0.05

**Table 2 medicina-58-01782-t002:** Characteristics of women with preferred CS as way of delivery.

Characteristics	Year 2010 *n* = 1175	Year 2020 *n* = 1033	*p*-Value
Higher socioeconomic status	36 (3%)	74 (7%)	<0.05
Higher education	97 (8.3%)	177 (17.1%)	<0.05
Medical education	12 (1.0%)	16 (1.6%)	<0.05
Nullipara	173 (14.7%)	242 (23.4%)	0.35
History of miscarriage	25 (2.1%)	59 (5.7%)	0.59
History of vaginal delivery	24 (2.0%)	33 (3.2%)	<0.05
History of cesarean section	58 (4.9%)	160 (15.5%)	<0.05

## Data Availability

The data presented in this study are available on request from the corresponding author. The data are not publicly available.

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
