# Peer review of "Factors Influencing Polish Women’s Preference for the Mode of Delivery and Shared-Decision Making: Has Anything Changed over the Last Decade?"

_medicina, 2022, doi:10.3390/medicina58121782_

Round 1
Reviewer 1 Report
The authors present factors influencing preference about the way of delivery. The number of data seems enough to explain the author's opinions. The data was clear and well-described. However, it is too difficult to recognize the difference between the percentages or numbers since there is no table or figure. Please provide figures or tables to describe the results. Then this manuscript becomes more attractive to the readers.
Author Response
Thank you for your remarks. We corrected the manuscript accordingly and provided tables to describe the results.
Reviewer 2 Report
Dear authors,
this article deals with an interesting topic. In my opinion, the results are confusing, there are many numbers in the text without statistical evaluation. I recommend completing at least two tables - see below. After revision, in my opinion, the article will be suitable for publication.
Another comments
lines 27 -28
" In 2010 34.9% of women with 27 a history of CS only,...."
Note: Abbreviation CS should be explained
lines 79 -80
"The exclusion criteria were not female gender, minority (less than 18 years old) ads missing of conflicting data."
Note: I do not understand this sentence.
Lines 92 - 93
"The approval from Warsaw Medical University Ethics Committee was obtained 19-03-2013 from code AKBE/21/13."
Note: Probably it is better: ..... was obtained on 19-03-2013 with code AKBE/21/13. ?????
Lines 107 - 128
"In 2010 VD was preferred by 66.0% of women with primary education, 69.0% with secondary education, 70.4% with higher education and 86.3% of those with medical education (p<0.05). In 2020, VD was preferred by 33.3% of women with primary education, 57.1% - secondary, 55.3% - higher and 69.3% - medical education (p<0.05).
Women were also asked about the decision-making process about the mode of delivery. Among the respondents with primary education, 18.9% in 2010 and 52.6% in 2020 (p-value is missing) thought women should have the independent right to decide about the mode of delivery, 58.5% vs 26.3% (p-value is missing) preferred shared decision-making with their obstetrician and 22.6% vs. 21.1% (p-value is missing) accepted CS only for medical indications. Independent right to decide about the mode of delivery was recognized by 32.1% in 2010 and 33.0% in 2020 (p-value is missing) of women with secondary education, 26.3% vs. 36.4% (p-value is missing) with higher education and 10.8% vs. 26.4% with medical education (p<0.05). Shared decision-making was the preference of 46.0% in 2010 and 44.0% in 2020 women with secondary education, 46.8% vs 46.2% with higher education and 38.7% with medical education (p<0.05). CS for medical indications only was chosen by 21.0% of women with secondary education in 2010 vs. 20.6% (p-value is missing) in 2020, 24.6% vs. 16.4% (p-value is missing) with higher education and 50.5% vs 36.8% - with medical education (p<0.05).
In 2010, 18.9% of women with primary education were willing to have CDMR, with respectively 27.4% - with secondary, 23.1% - with higher and 13.8% with medical education (p<0.05). Results from 2020 group did not have statistical significance. In 2010, 37.8% of women with primary education were willing to pay for CDMR, compared to 15.8% in 2020, respectively 40.1% vs. 36.7% with secondary education, 45.1% vs. 51.2% with higher education and 27.2% vs 39.1% with medical education (p<0.05 - Does it mean that all the differences are p<0.05 ? It should be mentioned.)."
lines 147 - 150
"In 2010, the proportion of women who accepted paid CDMR increased with the declared socioeconomic status, with 31.% in the low income group, 40.7% in medium income and 49.7% in the high income group (p-value is missing), but such correlation did not exist in 2020."
lines 153 - 155 + 161
"75.0% of respondents in 2010 and 71.4% (please, add p-value) in 2020 declared no comorbidities. 18.25% of women in 2010 and 9.03.% (please, add p-value) in 2020 were pregnant at the moment of filling in the questionnaire. 15.8% of women in 2010 had a history of miscarriage vs. 20.0% (add p-value) women in 2020"
Note: Please, do not start a sentence with a number. F.e. A total of 75.0% .... or Altogether 75.0% ... or Total 75.0% .....
Line 161
"51.4% of respondents in 2010 had a history of previous pregnancy vs. 77.1% in 2020" (please, add p-value)
Lines 167 - 177
"When we compared women who had had only a VD with who had never had deliveries, we found that in 2010 only 6.9% of those who had had VD preferred CS as a better way of delivery, compared to 18.7% of nulliparas (p<0.05); respectively in 2020 8.9% vs. 25.3% (p<0.05). In 2010, 18.6% of women who had a VD thought that it should women’s autonomic right to decide about the way of delivery in comparison with 29.9% (add p-value) of nulliparas, 43.8% vs.49.7% (add p-value) opted for shared decision-making with the obstetrician and 34.5% vs. 19.8% (add p-value) stated that CS should be performed only for medical indications; respectively 26.8% vs. 35.0%, 44.8% vs. 47.5% and 27.4% vs.16.0% in 2020 (p<0.05). In 2010, 26.9% of nulliparas wanted to have DMR, compared to 12.4% (add p-value) of women with a history of VD, compared to 34.4% vs. 19.1% (add p-value) in 2020. A history of VD was not found to influence the acceptance of paid CDMR."
Lines 178 - 198
You should add p-values to all differences.
Lines 243 - 244
"The highest HRQoL was found in the group of women who preferred and had VBAC."
Note: Abbreviation VBAC should be explained
Proposal of tables - adjust as you see fit
The basic characteristics of subjects
|
Characteristics |
|
Year 2010 n = 1175 |
Year 2020 n = 1033 |
p-value |
|
Age |
Median, mean |
Median, mean |
0.xx |
|
|
socioeconomic status - income |
low |
XX (X.X%) |
XX (X.X%) |
|
|
medium |
|
|
|
|
|
high |
|
|
|
|
|
education |
primary |
|
|
|
|
secondary |
|
|
|
|
|
higher |
|
|
|
|
|
medical education |
|
|
|
|
|
Nullipara |
|
|
|
|
|
Number of pregnancies |
Median, mean |
Median, mean |
|
|
|
history of miscarriage |
|
|
|
|
|
History of vaginal delivery |
|
|
|
|
|
history of CS |
|
|
|
|
|
history of VBAC |
|
|
|
|
|
smoking |
|
|
|
|
|
Alcohol use |
|
|
|
|
The characteristics of women with preferred CS as way of delivery
|
Characteristics |
Year 2010 n = 1175 |
Year 2020 n = 1033 |
p-value |
|
Age > 35 years |
XX (X.X%) |
XX (X.X%) |
0.xx |
|
higher socioeconomic status |
|
|
|
|
higher education |
|
|
|
|
medical education |
|
|
|
|
Nullipara |
|
|
|
|
history of miscarriage |
|
|
|
|
History of vaginal delivery |
|
|
|
|
history of CS |
|
|
|
|
history of VBAC |
|
|
|
Author Response
Dear Reviewer 2,
Thank you for your thorough analysis of our manuscript.
Following your recommendation, we completed the two tables you were kind to suggest, hopefully having ameliorated the presentation and visibility of the results.
We added the explanation for CS abbreviation in the Abstract.
We corrected the sentence about the exclusion criteria and removed the punctuation and spelling mistakes.
We corrected the sentence about the approval from Warsaw Medical University Ethics Committee.
Following your remarks, we added the p-value in all the lines of manuscript you had suggested in your review.
We changed the sentences that started with a number according to your suggestion.
We explained the abbreviation CBAC in lines 243-244.
Round 2
Reviewer 2 Report
Dear authors,
the article is now eligible for publication.
Another comments
Lines 96 - 98
"The approval from Warsaw Medical University Ethics Committee was obtained was obtained on 19-03-2013 with code AKBE/21/13 "
Note: There is written twice: was obtained